# Validity and Reliability of the New Basic Functional Assessment Protocol (BFA)

**DOI:** 10.3390/ijerph17134845

**Published:** 2020-07-05

**Authors:** Raquel Hernández-García, María Isabel Gil-López, David Martínez-Pozo, María Teresa Martínez-Romero, Alba Aparicio-Sarmiento, Antonio Cejudo, Pilar Sainz de Baranda, Chris Bishop

**Affiliations:** 1Department of Physical Activity and Sport, Faculty of Sports Sciences, Regional Campus of International Excellence “Campus Mare Nostrum”, University of Murcia, 30720 San Javier (Murcia), Spain; rhernandezgarcia@um.es (R.H.-G.); mariateresa.martinez13@um.es (M.T.M.-R.); alba.aparicio@um.es (A.A.-S.); psainzdebaranda@um.es (P.S.d.B.); 2Sports and Musculoskeletal System Research Group (RAQUIS), University of Murcia, 30100 C.P. Murcia, Spain; david.martinez.pozo@mail.ucv.es; 3London Sport Institute, Middlesex University, London NW4 4BT, UK; C.Bishop@mdx.ac.uk

**Keywords:** fundamental skills, basic motor pattern, quality of movement, functional assessment, qualitative analysis, content validity, inter-observer reliability

## Abstract

The global evaluation of motion patterns can examine the synchrony of neuromuscular control, range of motion, strength, resistance, balance and coordination needed to complete the movement. Visual assessments are commonly used to detect risk factors. However, it is essential to define standardized field-based tests that can evaluate with accuracy. The aims of the study were to design a protocol to evaluate fundamental motor patterns (FMP), and to analyze the validity and reliability of an instrument created to provide information about the quality of movement in FMP. Five tasks were selected: Overhead Squat (OHS); Hurdle Step (HS); Forward Step Down (FSD); Shoulder Mobility (SM); Active Stretching Leg Raise (ASLR). A list of variables was created for the evaluation of each task. Ten qualified judges assessed the validity of the instrument, while six external observers performed inter-intra reliability. The results show that the instrument is valid according to the experts’ opinion; however, the reliability shows values below those established. Thus, the instrument was considered unreliable, so it is recommended to repeat the reliability process by performing more training sessions for the external observers. The present study creates the basic functional assessment (BFA), a new protocol which comprises five tasks and an instrument to evaluate FMP.

## 1. Introduction

Assessments are among the main elements for practitioners to make informed and supported decisions for practice, which must be applied with documentary evidence in order to evaluate and improve sport performance [1]. The evaluation of human movement from an objective perspective, may have an impact on the learning and/or training process and, consequently, on performance [2]. In the last 10 years, there has been a change in the screening of musculoskeletal abilities, evolving from muscle and joint analysis towards a more integrated approach [3], in which the emphasis lies on the visual analysis of movement patterns during functional tasks [4].

According to Goodway, Gallahue and Ozmun [5], motor skills/abilities can be assessed from two points of view: product-oriented or process-oriented. Product-oriented evaluation implies focusing on the result of a movement; while process-oriented refers to the assessment of the movement execution. This latter type of evaluation process refers to the assessment of the quality of movement, which is defined as the movement that is executed with proper posture, breathing, mobility and coordination [6]-in short, movements or specific tasks performed in an optimal and efficient way [7]. It is also defined as the cognitive self-awareness of people expressed in global and local body functionality [8]. By means of the evaluation process of global movement patterns during functional tests, the necessary neuromuscular control synchrony, range of motion, strength, resistance, balance and coordination to complete movement can be examined [9]. In fact, these types of tests are being used in order to detect “poor” movement patterns, which in turn are related to a plausible injury risk [10] due to its possible variation of joint loading, strength, power and, of course, an accumulation of stress because of the lack of motor efficiency [7,11]. If this is associated with the following Ardern et al. statement, “the current rate of injured athletes are caused by functional shortage such as motor control and neuromuscular stability” [12], the use of tests that assess the quality of movement makes sense [13], especially to obtain results that help in diagnosis, evaluation and risk of injury and training control [14,15]. In addition to recognizing motor problems, this assessment evaluates interventions and predicts the recovery of people who have suffered an injury [16,17].

Movement patterns are basic sequences programmed in the central nervous system, which respond to global movement, where the human body organizes itself to achieve an intention [18,19]. That is why the tests used to evaluate the quality of movement are posed through global and simple tests, where the dynamic behavior of the individual is observed and analyzed [20]. There are many tools to assess movement [5]-numerous prior studies have carried out laboratory evaluations based on 3D kinematic and kinetic task analysis [21]-nevertheless, these types of tests require expensive and sometimes inaccessible equipment; thus, determining more affordable ways to examine movement quality poses advantages for practitioners [22]. With this in mind, observational methodology should be trusted, that is, the process of compiling, organizing and giving sense to the visual, aural and sensory information that is obtained from a person in motion [23]. When dealing with motor function, focusing mainly on visual information gathered from human performance will be necessary [24]. A close link has been shown between the quality of movement and the observational methodology in order to obtain qualitative information [25]. Therefore, as different protocols have been created, so has the assessment of the quality of movement that observational methodology uses [7].

Within the most well-known motion detection protocols by the scientific community, it is possible to find the most utilized ones: (a) Movement Competency Screen (MCS) [26], composed of six tests (posture, squat, lung-and-twist, push-up, bend-pull and single leg squat), was developed in an attempt to be employed among sport and health professionals, providing them with a better comprehension of the athlete’s movement ability before the prescription of a strength training program. Nevertheless, research has shown some contraindications, such as a deficient reliability among juvenile athletes [27] and weak associations between the total MCS score and injury risk [28]. In addition, null results of the evaluation of the asymmetry using MCS have been found, given the importance some authors show in the assessment of asymmetry [29,30]. Therefore, a battery which takes this into account is needed. (b) Athletic Ability Assessment (AAA) [31], which is composed of seven tests (prono hold, side hold, overhead squat, single leg squat, walking lunge, single leg hops and lateral bound). McKeown backs up the idea that the exercises used within their protocol are closely linked to the basic abilities that support sport performance. This battery was created with the purpose of being employed as an assessment tool of athletic profiles and to evaluate the changes in functional ability throughout time. However, the AAA battery has lower reliability among less experienced athletes [32]. (c) Functional Movement System (FMS) [4,11], which consists of seven tests (Deep squat, hurdle step, in-line lunge, shoulder mobility, active straight leg raise, trunk stability push-up and rotary stability). Initially, it was created to assess athletes who were trying to reach their highest athletic performance. Later on, it was established as a protocol in order to detect weak motor patterns that generate compensations, and its use was considered suitable among people who were training for health purposes. FMS is one of the most frequently used tools for an individual’s functional assessment. However, it generates opposition, with different authors indicating poor reliability and validity [33], and an inability to predict injury risk [34]. Other authors question the battery punctuation system and suggest that the punctuation method used in FMS does not provide the trainer with specific data about the subject’s functionality [35,36,37].

All of these tests contain, in a detailed way, a register of the actions that need to be observed. This register has gone through a process of ratification and reliability in order to verify that it fulfils the aim for which it has been designed, and it is essential that all the tests accomplish this requirement [38]. In this sense, the need to design a test battery, which is capable of assessing fundamental motor patterns (FMP), appears, which is defined as the base and the essence of more complex movements [39] through observational methodology in any type of individual.

Therefore, the aims of the present study were: (a) to design an easy, simple and concise protocol that can be carried out in a field-based context to provide sport science experts with information about the quality of movement in fundamental motor patterns that can be executed by any individual; (b) to analyze the validity and reliability of the instrument created for the analysis of these fundamental patterns of motion.

## 2. Materials and Methods

### 2.1. Design

This study was designed in order to examine the validity and reliability of a new protocol that assesses basic functionality in five different movement patterns. First of all, the instrument was created. The study began with the selection of five tasks that would serve for the evaluation of different basic patterns of motion; all of them have been already studied in the scientific: Over Head Squat (OHS) [40]; Hurdle Step (HS) [11]; Forward Step Down (FSD) [41]; Shoulder Mobility (SM) [42] and Active Stretching Leg Raise (ASLR) [42] (Appendix A). The test selection criteria were the following: (a) having bibliographical backing available as an individual task; (b) it responds from the human motion perspective as FMP, that is, it is the basis of more complex movements, permitting the possibility of execution without materials; (c) its combined performance may provide us with information about the individual’s global functional condition.

Once the five tests had been established, a list of compensations that the body can produce when performing each of the established movements was created. The list of compensations that had to be detected in each task was developed, taking into account the scientific literature that relates injury risk with incorrect motor manifestations (for instance, knee valgus, pelvis side inclination or similar manifestations). Once the list of manifestations for observation was proposed, during the second stage of the process, ten qualified judges were asked to make a qualitative and quantitative assessment of the instrument. In order to do that, a document including all the proposed variables for each of the tasks was elaborated, as shown in Table 1, which contained three sections per variable for the judges to evaluate: definition (description of the items); belonging degree (if the variable is considered suitable or not for its inclusion inside the task); information collection and punctuation (if it is considered to be opportune or not in the punctuation system). The screening was accomplished by means of a Likert type quantitative scale from 1 to 10, and extra space was also provided in case the general qualitative evaluation of any element was required.

Judges were given a 15 day deadline for completing the document. Using the information gathered from the qualified experts during their first review, some items were modified in their definition as they did not fulfil the minimum value required and some adjustments were conducted, such as the inclusion of the two new items in the suggested categories by the different experts. At the third phase of the assessment process, a new report was written and the same group of qualified judges was asked to elaborate a qualitative and quantitative evaluation of the different items. At this stage, the purpose was to verify the degree of belonging of the object of study and the level of accuracy of the different categories and items. A 15 day deadline for filling out the document was established again. Subsequently, the internal assessment was calculated once again using the data that were extracted after the second revision was completed by the qualified judges. After their second analysis, the list of variables remained permanently established and closed.

At the fourth stage, the reliability of the tools was calculated. This process required the training of six observers that were external to the validation process, for this was held as a training session and two sessions where the reliability study was planned-each of the sessions were 120 min each. The first session was focused on explaining the categories and their codification to let the observers become familiar with them, and then on carrying out a training class for the observers through real situations. At the second session, each observer completed the Basic Functional Assessment (BFA) with the same case, which was redone at the third session 15 days later. The Kappa of Cohen index was calculated for inter-observer and intra-observer reliability of the tool. Reliability within the same observer or intra-rater reliability (a case studied by the same observer in different occasions), was calculated by means of correlating the results from the first observation (at the second session) with those from the second observation (at the third session). Reliability among observers (inter-rater reliability) was estimated by correlating the results from observer 1 with the results from observers 2, 3, 4, 5 and 6, successively with all six participating observers. This process among the observers was completed with the results both from the first and the second monitoring sessions, respectively.

At the fifth and final phase of the assessment process, taking into account all the collected data from the previous stages and doing the necessary adjustments, the process was finalized with the creation of the basic functional assessment (BFA) as an observational tool to evaluate FMP, designed so that it can be undertaken in a minimal amount of time and without the need of using expensive materials. A great step on injury prevention is to concentrate the existent resources in large towns where there are limited amounts of time and resources, as it is in the case of juvenile leagues, state school systems, minor sports clubs, etc. [43].

### 2.2. Participants

Firstly, a total of ten qualified judges, four women and six men, aged between 30–38 years old, participated in the study. All of them were experts in the matter, with a minimum experience in functional assessment of ten years. Although the majority were doctors of Sport Sciences, three guidelines were established, from which at least one had to be fulfilled: (a) to have a degree in Sport Sciences; (b) to have more than five years of experience in functional evaluation both for sportspeople and for non-sportspeople; (c) be currently active in the training/physical therapy professional environment. Secondly, six external observers to the process of validation-two women and four men-between the ages of 21–25 years old, were trained to perform the BFA and participated in the study of reliability. Three of them were students of the last course of the Grade of Sciences of Sport and the other three had already finished the grade and they were students of a Master’s degree of investigation in Sciences of Sport. The inclusion criteria for the external observers were: (1) to be a student or to have finished the Grade of Sport Sciences; (2) to attend the training class that took place during the first session.

### 2.3. Variables

In order to have the observational tool elaborated and assessed by the eight experts, two types of variables were essential-calculation variables and the categorical variables.

For the evaluation of the content, the qualified judges estimated the value of the “Belonging” and “Definition” sections from each variable by means of a Likert type quantitative scale from 1 to 10. In case it was necessary, there was also an additional section for a possible general qualitative assessment of each item available. When referring to the categorical variables, each movement involved in the execution of every proposed functional test was taken as a unit of measurement, taking into account three different aspects: the plane from where the observation should take place (sagittal plane, anterior frontal plane and posterior frontal plane); the body area that is observed, which is composed by several items making reference to different body areas (thorax, femur, knee, foot, hip, pelvis, arms, lumbar, cervical, etc.); the type of movements that can be observed and what provides the information about compensations being compensation movements (external and internal rotation, pelvic tilt, valgus, varus, heel lift, etc.). These movements were assessed in both the right and left extremities.

### 2.4. Statistical Analysis

So as to get the validity of the observational tool through the process that was carried out by qualified judges, the validity of content index was determined by calculating the coefficient of Aiken’s V [44] using the following equation: $V=\frac{x-l}{k}$.

This equation takes into account the number of items, the number of judges, just like the range of assessment for each item; allowing us to check if the obtained magnitude is optimal in terms of content validity in the different items. X is the mean of the judges’ marks, *l* is the minimum scale score and *k* is the scale range that was used. In order to reject the void hypothesis (V0), the significance level was 0.69. Items whose mean values were below 0.69 were eliminated. The items that had mean values between 0.69 and 0.80 were modified, while those items whose values were above 0.80 did not change.

For the reliability analysis, Cohen’s Kappa value [45] was used, the values were classified using the following criteria: trivial (0.1), small (0.1–0.3), moderate (0.3–0.5), large (0.5–0.7), very large (0.7–0.9) or practically perfect (0.9). The statistical analysis was performed using IBM SPSS Statistics for Windows (version 24.0) (IBM Corp, Armonk, NY, USA).

## 3. Results

Table 2 shows the final compensations reviewed by the expert judges for each of the test. Finally, after the modifications of some of the variables, such as the inclusion of cervical flexion/extension in the OHST test, suggested by the experts. The battery remained composed in the following way: OHST was composed of a total 15 compensations, HST and FSDT match coincided with a total of 17 compensations, SMT test 3 types of compensations and finally ASLR with 10 compensations. All of them are evaluated bilaterally, clearly differentiating whether compensation is made in the right or left hemisphere of the body or on both sides.

The planes from which the different offsets were observed are also established. OHST, HST and FSDT are observed from three planes-front, sagittal and back. SMT is observed from the sagittal plane and back, and ASLR is observed only from the sagittal plane. The images detail which points we need to look at to detect the compensation.

Table 3 shows the results of the validity of the variables after the second review of the expert judges. The table shows the validity data for the OHS task where all of the items obtained optimal values in terms of belonging for its inclusion; however the item, excess thoracic kyphosis, did not obtain a value suitable for its definition and should be improved.

Table 4 shows the validity data for the HS task-all the items obtained optimal values both in their definition and in their belonging.

In Table 5 we find the validity data for the FSD task-all the items obtained optimal values both in their definition and in belonging.

The data in Table 6 refer to the SM task-all their items obtained optimal values regarding belonging; however, the data show that definitions are not optimal and should be improved for its inclusion.

Finally, Table 7 shows the data of the variables corresponding the ASLR task-all these variables obtained optimal values in terms of their belonging and their definition.

Bearing in mind the results of Table 3 and Table 6, in Table 8 shows a new definition is proposed for variables that have not met the minimum value. These variables have obtained the maximum value in terms of belonging, so we considered that they should be included in the observation sheet, modifying for its definition.

Table 9 shows the results of intra-observer reliability, calculated on each test and together as a single battery. The results obtained are dispersed, the SM task being the only one that obtains a practically perfect reliability with a Kappa value of 1 in the three observers, while the contrary occurs with the FSD task, where a small reliability appears (0.1–0.3). We also found a trivial value (0.1) in the HS task. The reliability data of the battery as a whole are also shown, where only one of the observers attained very large reliability (0.75).

The values of inter-observer reliability were made only from the (BFA) battery and at both observation periods, showing different values in each period (Table 10). The first observation obtained moderate-small reliability (0.1–0.5) and the second observation obtained small reliability (0.1–0.3). Some large inter-observer reliability values were obtained (0.5–0.7), one in the first observation and two in the second observation. Finally, the BFA battery shows a intra–inter observer reliability of moderate-small.

## 4. Discussion

The objectives of the present study were to design and to analyze the validity and the reliability of an observational sheet, aimed at drafting a Basic Functional Assessment battery (Table A1) which is able to provide us with information about the quality of movement in PMF. The results showed that the BFA is considered valid for its use in detecting alterations in PMF. It also showed a low to moderate intra and inter-observer reliability for BFA. Kappa values presented higher values in isolated tests in terms of intra-observer reliability.

Table 2 exhibits the final manifestations that were revised by the expert judges for each of the tests. A total of 61 manifestations were established, of which 53 of them must be evaluated in a unilateral way. Body asymmetry may be associated with a higher presence of injuries [46,47]. Some authors report the importance of asymmetry assessment in their studies [20,29,35,48].

Some proposed modifications by the experts were made, such as including cervical extension and flexion. This compensation has been highlighted within the back squat correct technique [10,49]. The group of experts considered keeping a great number of the variables and the proposed definitions. Some of these variables are studied in other tasks (e.g., knee valgus/varus, heel lift, foot pronation/supination, lumbar kyphosis/lordosis, pelvic tilt). Kritz [26] uses them especially when assessing the following tasks: squat; lunge and twist; single leg squat. Similarly, so do Bennett et al. [50], who employ these variables particularly in squat and overhead reach test evaluations. In other studies, the use of other variables that have also been accepted by the experts are found, such as the manifestation of the arms falling forward, utilized within the overhead assessment [35,50]. Padua et al. [43] apply the internal/external rotation of the feet for the jump-landing task evaluation as an injury predictor. Myer et al. [49] describe, in a similar way, the pelvic tilt during back squat. Park et al. [41] use the FSDT test, and its assessment is based on the criteria that coincide with some of those suggested (e.g., torso movement, aligned knees, pelvic tilt and rotation) and they also bear in mind the external pelvic rotation. However, as opposed to how it is contemplated in BFA, this last variable is evaluated by means of using a dynamometer. In Cook et al. [4], for FMS battery during an ASLR test, some advice for their execution are proposed that coincide with some of the suggested variables in this test-the external rotation of the leg that remains on the ground and laid out knees.

Nevertheless, not all definitions were accepted by the experts. In the OHST test, the manifestation-an excess in thoracic kyphosis-seen from the sagittal plane, did not get a valid definition. Myer et al. [49] emphasize this manifestation and claim that the thoracic spine should preferably be extended and rigid. In the case that it is not able to stand, it may suggest weakness in the spinal erectors, trapezius and rhomboid, as well as an upper crossed syndrome. Due to the fact that its inclusion within the test has been considered accurate and valid and it is considered an important manifestation in order to include it in the OHST evaluation, since it may provide with valuable information about poor PMF, a new definition has been proposed, as shown in Table 4. In the SMT test, all the manifestations-winged scapula, lordosis excess and cervical protraction-obtained excellent belonging validity; however, their definitions were not suitable. In spite of this, the SMT test was considered to be included within the BFA. In a study published by Larsen et al. [51] it was corroborated that simple visual observation methods to assess the scapular function present a better reliability compared with other types of more complex measuring. The SMT test is used to evaluate the functionality of the upper part of the body [52]. Manifestations proposed for SMT are considered suitable to discover poor motion patterns within the upper extremities. The presence of winged scapulars could provide us with information about the existence of other alterations that may limit the scapula-humeral functionality. In a study of cases which were published by Martínez Bermudez et al. [53], it was found that all the Parsonage-Turner syndrome cases showed the presence of winged scapula. Other studies reported the same data [54,55]. Concerning the manifestation of lordosis excess, Kritz [26] points out the importance of debating the role of the lumbar area in upper-body movement tasks, since the lumbar area is responsible for stabilizing the spine during upper-body movement tasks. If there is not proper lumbar stabilization, the needed strength for the shoulder to work may be compromised [56]. Since we are dealing with manifestations that may provide us with valuable information about poor PMF in an upper member, new definitions have been proposed, as shown in Table 4. All the suggested definitions in Table 4 must be re-evaluated by the experts, so as to obtain a validity value in the suitable definition and, therefore, include them within the BFA protocol.

Another objective of this study was to test the intra-and inter-observer reliability of the BFA. Intra-observer reliability was conducted individually for each test and combined as a BFA battery; meanwhile inter-observer reliability was directed only in a combined way.

All the values were considered inferior to those that were seen as suitable, showing a BFA low-to-medium reliability. Although a specific limit to determine if the reliability coefficient is high or not does not exist, a coefficient higher than 0.70 is considered acceptable [57], reaching this value on very few occasions. In other studies, in which observational methodology as an assessment system is used, close values to those obtained ones found. Rogers et al. [58] obtained a deficient intra/inter-observer reliability in the AAA battery when carrying it out among Australian sub-elite football players. Weir et al. [59], following bibliographical recommendations, chose six core stability evaluation tests and showed a deficient intra/inter-observer reliability. In the same line, we found Inovero et al. [27], who also displayed a deficient intra/inter-observer reliability among university volleyball athletes. Dekkers et al. [60] analyzed the Observable Movement Quality scale reliability among children between the age of six months and six years, and they obtained moderate inter-observer reliability results.

The results obtained in BFA reliability may be due to certain limitations that the study exhibit, such as the fact that the observers were Sport students only, who had little experience with this type of methodology. Facing this limitation, it is interesting to establish strategies to improve BFA reliability, such as: (1) Increasing the training among the observers. Inovero et al. [27] conducted a validation process for the MSC test by only carrying out two formative sessions and obtained similar results to ours with a low reliability. In this manner, Rogers et al. [58] executed a reliability process for the AAA battery, having two formative hours and obtaining low reliability results. It is likely that a coordinated and standardized formation may help improve the utility of the system among evaluators [36,61]. (2) Carrying out the study relying on the expert and novice observers’ collaboration. The different values obtained in each of the individual tests could be due to the familiarity and experience that the observers have in each of the tests. In a study published by Bennett et al. [50], the impact of the evaluator’s experience within the reliability data in a Movement Screen battery was determined. High reliability data were obtained among evaluators, decreasing these values among the novice evaluators and also pointing out the importance of a standardized formation for inexperienced evaluators, since the learning associated with the movement quality assessment leads to a more consistent punctuation [7]. Weeks et al. [62] also showed the importance of the observer’s experience as, in their study they obtained higher reliability data than those who had more experience.

On the other hand, the observational sheet was designed with a punctuation system in which the BFA maximum result may reach 76. If the subject manifests a compensation it will count as “1 point”, if it does not manifest any compensations it will count as “0 points” and it quantifies as the addition of: 14 possible errors of individual movement for the OHST; 22 for the HST; 22 for the FSDT; 6 for the SMT; 12 for the ASLRT. Each of the tests is marked individually and, joining all of them, the global punctuation is obtained for the person’s motion quality. This punctuation system has certain advantages, since it allows us to perform comparatives during the re-evaluation, as well as permitting the professionals to execute data analysis about the movement quality and other aspects about the athlete’s physical development [63,64]. The use of a numerical marking system has been debated among authors. During a systematic revision [2] it was shown that the general punctuation of poor quality of movement is associated with a higher risk of injury in the lower extremities. Mann et al. [65] demonstrate that a total punctuation may be used in a more reliable way than an individual one when assessing movement. Despite the fact that the global punctuation system has been used in research, there is scientific evidence that contradicts this marking system [33]. Bonazza et al. [37], in a systematic revision about FMS, show a low inner validity in systems that employ a numerical marking system, and declare that the results must focus on individual punctuation instead of a global mark. Kazman et al. [66] maintain the idea that, when employing the numerical marking system, every test must be graded as a one-dimensional construction. Between both perspectives lie O’Connor et al. [36], who do not recommend the use of a general numerical marking system as the only risk of injury identification method.

The study has some limitations, such as the lack of training sessions for the reliability process. It is likely that, by performing a more extensive training, observers will be more familiar with the BFA assessment process and obtain better reliability values of the instrument. Therefore, it is considered of importance to repeat the intra and inter-observer reliability process, persevering previously commented characteristics since a reliable observational tool among observers is an important aspect. Having consistency among observers indicates that the different people may employ the instrument and obtain similar results [35].

Results suggest that BFA may have potential to establish motion quality in different subjects. For future research lines, it is recommended to investigate reliability and open other lines, such as their use in the detection of injury risk among amateur sportspeople, seeing as it consists of a movement quality assessment that does not require specific tools from a laboratory. This is also interesting for its application in sport centers as an initial observational form to establish different action protocols, allowing for the individualization of the user during training.

## 5. Conclusions

The present study creates the basic functional assessment (BFA), a new protocol which comprises five task and an instrument to evaluate a selection of five fundamental motor patterns (FMP). The BFA has been designed to be an easy, simple and concise protocol that can be carried out in a field-based context, to provide sport science experts with information about the quality of movement in FMP that can be executed by any individual and are the basis to develop more complex movements.

The previously described tool is considered as valid so it is able to accept its use as a tool for the Basic Functional Assessment. It has been shown that the tool is not reliable in its measure. This study has some limitations, such as the lack of training sessions of observers for the reliability process. Therefore, the reliability process must be repeated, taking into account the limitations of the study. This process can be considered as a future line of research, as well as working on other lines, such as using BFA in detecting the risk of injury and obtaining battery data based on the field.

## Figures and Tables

**Table 1 ijerph-17-04845-t001:** Example questionnaire for expert judges.

**External Rotation Support Foot Right; Front Plane Anterior**											
Definition: Turning the foot on the longitudinal axis until the phalanges of the second right/left toe are oriented in a lateral direction											
Poorly defined	1	2	3	4	5	6	7	8	9	10	Well defined
**Proposed Definition, in Case the Anterior does not Remain Clear**											
Accuracy: Do you think the inclusion of this variable within an instrument for assessing the compensation dimension in this test is relevant?											
Not pertinent	1	2	3	4	5	6	7	8	9	10	Very pertinent
Scoring: 1 point will be awarded if the subject manifests this variable, 0 points if they do not manifest. Do you think this score is appropriate to assess the compensations that can be manifested in HST?											
Not adequate	1	2	3	4	5	6	7	8	9	10	Very adequate

**Table 2 ijerph-17-04845-t002:** Total compensations after review by expert.

View	OHS							
**Front Plane (FP)**	External rotation foot Left (L)/Right (R) *		Internal rotation foot L/R *		Valgus knee L/R *		Varus knee L/R *	
	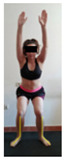		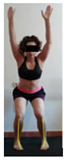		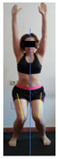		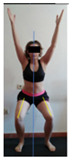	
**Back Plane (BP)**	Thorax rotation *		Foot pronation L/R *	Foot supination L/R *	Asymmetrical distribution of the hip to the L/R *		Heel lift *	
	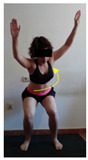		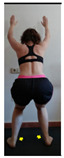	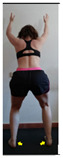	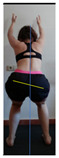		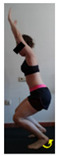	
**Saggital Plane (SP)**	Lumbo–pelvis dissociation loss *	Excess lumbar lordosis *	Excess thoracic *	Arms fall to the front *	Cervical flexion **		Cervical extension **	
	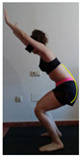	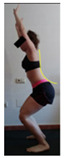	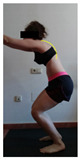	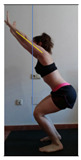	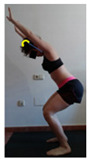		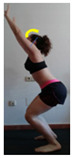	
**HS**								
**FP**	External rotation support foot L/R *	Internal rotation support foot L/R *	Valgus support knee L/R *	Varus support knee L/R *	External rotation Hip L/R flexed *		Internal rotation Hip L/R flexed *	
	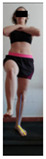	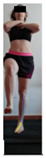	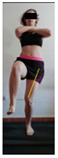	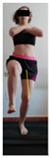	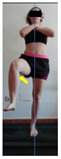		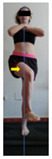	
	Pelvis tilt L/R *	Pelvis rotation L/R *	Thorax rotation towards the hip in flexion L/R *		Thorax rotation opposite hip in flexion L/R *		Thorax movement *	
	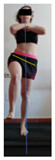	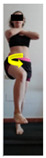	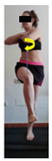		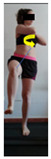		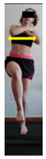	
**BP**	Support foot pronation L/R *		Support foot supination L/R *					
	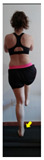		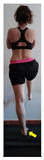					
**SP**	Heels lift, support foot L/R *	Lumbo–pelvis dissociation loss, the leg L/R * supported		Excess lumbar lordosis, the leg L/R * supported		Excess thoracic kyphosis, the leg L/R * supported		
	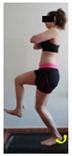	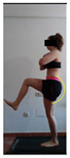		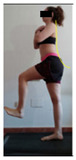		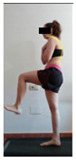		
**FSD**								
**FP**	External rotation support foot L/R *	Internal rotation support foot L/R *	Valgus support knee L/R *	Varus support knee L/R *	External rotation extended leg *		Internal rotation extended leg *	
	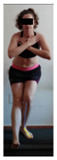	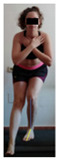	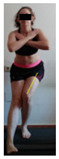	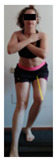	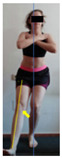		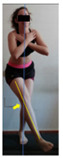	
	Pelvis tilt L/R *	Pelvis rotation L/R *	Thorax rotation towards the leg supported *		Thorax rotation opposite the leg supported *		Thorax movement *	
	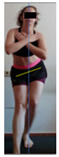	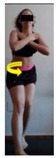	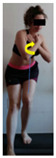		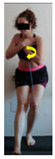		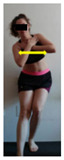	
**BP**	Support foot pronation L/R *		Support foot supination L/R *					
	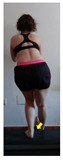		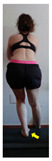					
**SP**	Heels lift, support foot L/R *	Lumbo–pelvis dissociation loss, the leg L/R * supported		Excess lumbar lordosis, the leg L/R * supported		Excess thoracic kyphosis, the leg L/R * supported		
	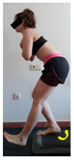	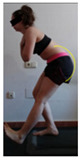		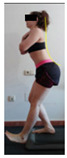		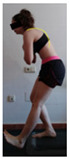		
**SM**								
**BP**	Winged scapula, internal rotation arm L/R *	**SP**	Excess lumbar lordosis, internal rotation arm		Cervical protraction, internal rotation arm L/R *			
	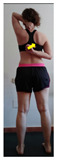		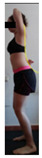		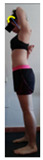			
**ASLR**								
**SP**	External rotation, extended hip L/R *		Internal rotation, extended hip L/R		Extended leg modification L/R *		Modification of the raised leg L/R *	
	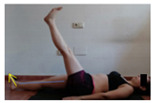		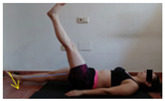		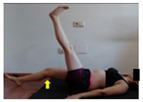		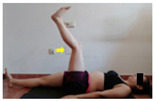	
	Flexion thoracic, hip flexion L/R *		Extension thoracic, hip flexion L/R *		Flexion lumbar thoracic, hip flexion L/R *		Extension lumbar thoracic, hip flexion L/R *	
	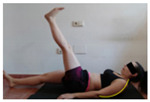		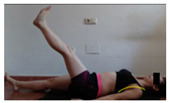		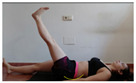		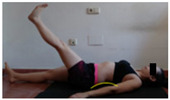	
	Extension cervical, thoracic, hip flexion L/R *				Flexion cervical, thoracic, hip flexion L/R *			
	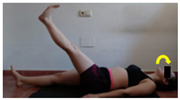				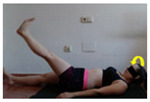			

LEGEND: OHS-overhead squat, HS-hurdle step, FSD-forward step-down, SM-shoulder mobility, ASLR-active straight leg raise, FP-Front plane, BP-Back plane, SP-Sagittal plane, L-Left, R-Right, * Compensation suggested by the bibliographic review, ** Compensations suggested by experts.

**Table 3 ijerph-17-04845-t003:** Assessment of Aiken V by ten experts of the proposed variables for OHS.

Variables	Definition	Membership
OHS		
External Rotation Foot Right; Front Plane Anterior	1	1
External Rotation Foot Left; Front Plane Anterior	1	1
Internal Rotation Foot Right; Front Plane Anterior	1	1
External Rotation Foot Right; Front Plane Anterior	1	1
Internal Rotation Foot Left; Front Plane Anterior	1	1
Valgus Knee Right; Front Plane Anterior	1	1
Valgus Knee Left; Front Plane Anterior	1	1
Valgus Knee Left; Front Plane Anterior	1	1
Varus Knee Right; Front Plane Anterior	1	1
Varus Knee Left; Front Plane Anterior	1	1
Thorax Rotation to the Right; Front Plane Anterior	1	1
Thorax Rotation to the Left; Front Plane Anterior	1	1
Pronation Foot Right; Back Plane	1	1
Pronation Foot Left; Back Plane	1	1
Supination Foot Right; Back Plane	1	1
Supination Foot Left; Back Plane	1	1
Asymmetrical Distribution of the Hip to the Right; Back Plane	1	1
Asymmetrical Distribution of the Hip to the Left; Back Plane	1	1
Heels Lift; Sagittal Plane	1	1
Lumbo–Pelvis Dissociation Loss < 45°; Sagittal Plane	1	1
Excess Lumbar Lordosis; Sagittal Plane	1	1
Excess Thoracic Kyphosis; Sagittal Plane	0.33	1
Arms Fall to the Front; Sagittal Plane	1	1
Cervical Extension; Sagittal Plane	1	1
Cervical Flexion; Sagittal Plane	1	1

**Table 4 ijerph-17-04845-t004:** Assessment of Aiken V by ten experts of the proposed variables for HS.

Variables	Definition	Membership
HS		
External Rotation Support Foot Right; Front Plane Anterior	1	1
External Rotation Support Foot Left; Front Plane Anterior	1	1
Internal Rotation Support Foot Right; Front Plane Anterior	1	1
Internal Rotation Support Foot Left; Front Plane Anterior	1	1
Valgus Support Knee Right; Front Plane Anterior	1	1
Valgus Support Knee Left; Front Plane Anterior	1	1
Varus Support Knee Right; Front Plane Anterior	1	1
Varus Support Knee Left; Front Plane Anterior	1	1
External Rotation Hip Right Flexed; Front Plane Anterior	1	1
External Rotation Hip Left Flexed; Front Plane Anterior	1	1
Internal Rotation Hip Right Flexed; Front Plane Anterior	1	1
Internal Rotation Hip Left Flexed; Front Plane Anterior	1	1
Pelvis Tilt, Hip Right Flexed; Front Plane Anterior	1	1
Pelvis Tilt, Hip Left Flexed; Front Plane Anterior	1	1
Pelvis Rotation, Hip Right Flexed; Front Plane Anterior	1	1
Pelvis Rotation, Hip Left Flexed; Front Plane Anterior	1	1
Thorax Rotation Towards Right, the Hip Right in Flexion; Front Plane Anterior	1	1
Thorax Rotation Towards Right, the Hip Left in Flexion; Front Plane Anterior	1	1
Thorax Rotation Towards Left, the Hip Right in Flexion; Front Plane Anterior	1	1
Thorax Rotation Towards Left, the Hip Left in Flexion; Front Plane Anterior	1	1
Thorax Movement, Hip Right Flexed; Front Plane Anterior	1	1
Thorax Movement, Hip Left Flexed; Front Plane Anterior	1	1
Pronation Right Foot in Support; Back Plane	1	1
Pronation Left Foot Supported; Back Plane	1	1
Supination Right Foot Supported; Back Plane	1	1
Supination Left Foot Supported; Back Plane	1	1
Heels Lift Right, Hip Left Flexed; Sagittal Plane	1	1
Heels Lift Left, Hip Right Flexed; Sagittal Plane	1	1
Lumbo–Pelvis Dissociation Loss < 45°, Hip Right Flexed; Sagittal Plane	1	1
Lumbo–Pelvis Dissociation Loss < 45°, Hip Left Flexed; Sagittal Plane	1	1
Excess Lumbar Lordosis, Hip Right Flexed; Sagittal Plane	1	1
Excess Lumbar Lordosis, Hip Left Flexed; Sagittal Plane	1	1
Excess Thoracic Kyphosis, Hip Right Flexed; Sagittal Plane	1	1
Excess Thoracic Kyphosis, Hip Left Flexed; Sagittal Plane	1	1

**Table 5 ijerph-17-04845-t005:** Assessment of Aiken V by ten experts of the proposed variables for FSD.

Variables	Definition	Membership
FSD		
External Rotation Support Foot Right; Front Plane Anterior	1	1
External Rotation Support Foot Left; Front Plane Anterior	1	1
Internal Rotation Support Foot Right; Front Plane Anterior	1	1
Internal Rotation Support Foot Left; Front Plane Anterior	1	1
Valgus Support Knee Right; Front Plane Anterior	1	1
Valgus Support Knee Left; Front Plane Anterior	1	1
Varus Support Knee Right; Front Plane Anterior	1	1
Varus Support Knee Left; Front Plane Anterior	1	1
External Rotation Right Leg, Left Leg Support; Front Plane Anterior	1	1
External Rotation Left Leg, Right Leg Support; Front Plane Anterior	1	1
Internal Rotation Right Leg, Left Leg Support; Front Plane Anterior	1	1
Internal Rotation Left Leg, Right Leg Support; Front Plane Anterior	1	1
Pelvis Tilt, Leg Right Support; Front Plane Anterior	1	1
Pelvis Tilt, Leg Left Support; Front Plane Anterior	1	1
Pelvis Rotation, Leg Right Support; Front Plane Anterior	1	1
Pelvis Rotation, Leg Left Support; Front Plane Anterior	1	1
Thorax Rotation Towards Right, The Leg Right in Support; Front Plane Anterior	1	1
Thorax Rotation Towards Right, The Leg Left in Support; Front Plane Anterior	1	1
Thorax Rotation Towards Left, The Leg Right in Support; Front Plane Anterior	1	1
Thorax Rotation Towards Left, The Leg Left in Support; Front Plane Anterior	1	1
Thorax Movement, Leg Right in Support; Front Plane Anterior	1	1
Thorax Movement, Leg Left in Support; Front Plane Anterior	1	1
Pronation Right Foot in Support; Back Plane	1	1
Pronation Left Foot Supported; Back Plane	1	1
Supination Right Foot Supported; Back Plane	1	1
Supination Left Foot Supported; Back Plane	1	1
Heels Lift Right, Leg Left in Support; Sagittal Plane	1	1
Heels Lift Left, Leg Right in Support; Sagittal Plane	1	1
Lumbo–Pelvis Dissociation Loss < 45°, Leg Right in Support; Sagittal Plane	1	1
Lumbo–Pelvis Dissociation Loss < 45°, Leg Left in Support; Sagittal Plane	1	1
Excess Lumbar Lordosis, Leg Right in Support; Sagittal Plane	1	1
Excess Lumbar Lordosis, Leg Left in Support; Sagittal Plane	1	1
Excess Thoracic Kyphosis, Leg Right in Support; Sagittal Plane	1	1
Excess Thoracic Kyphosis, Leg Left in Support; Sagittal Plane	1	1

**Table 6 ijerph-17-04845-t006:** Assessment of Aiken V by ten experts of the proposed variables for SM.

Variables	Definition	Membership
SM		
Winged Scapula, Right Arm Flexion and Left External Shoulder Rotation; Back Plane	0.5	1
Winged Scapula, Left Arm Flexion and Right External Shoulder Rotation; Back Plane	0.5	1
Excess Lumbar Lordosis, Right Arm Flexion and Left External Shoulder Rotation; Sagittal Plane	0.5	1
Excess Lumbar Lordosis, Left Arm Flexion and Right External Shoulder Rotation; Sagittal Plane	0.33	1
Cervical Protraction, Right Arm Flexion and Left External Shoulder Rotation; Sagittal Plane	0.33	1
Cervical Protraction, Left Arm Flexion and Right External Shoulder Rotation; Sagittal Plane	0.33	1

**Table 7 ijerph-17-04845-t007:** Assessment of Aiken V by ten experts of the proposed variables for ASLR.

Variables	Definition	Membership
ASLR		
Flexion Hip Right, Leg Left External Rotation; Sagittal Plane	1	1
Flexion Hip Left, Leg Right External Rotation; Sagittal Plane	1	1
Flexion Hip Right, Leg Left Internal Rotation; Sagittal Plane	1	1
Flexion Hip Left, Leg Right Internal Rotation; Sagittal Plane	1	1
Flexion Hip Right, Left Leg Support is Modified; Sagittal Plane	1	1
Flexion Hip Left, Right Leg Support is Modified; Sagittal Plane	1	1
Flexion Hip Right, Flexion Leg; Sagittal Plane	1	1
Flexion Hip Left, Flexion Leg; Sagittal Plane	1	1
Flexion Hip Right, Thorax Extension; Sagittal Plane	1	1
Flexion Hip Left, Thorax Extension; Sagittal Plane	1	1
Flexion Hip Right, Thorax Flexion; Sagittal Plane	1	1
Flexion Hip Left, Thorax Flexion; Sagittal Plane	1	1
Flexion Hip Right, Lumbar Extension; Sagittal Plane	1	1
Flexion Hip Left, Lumbar Extension; Sagittal Plane	1	1
Flexion Hip Right, Lumbar Flexion; Sagittal Plane	1	1
Flexion Hip Left, Lumbar Flexion; Sagittal Plane	1	1
Flexion Hip Right, Cervical Extension; Sagittal Plane	1	1
Flexion Hip Left, Cervical Extension; Sagittal Plane	1	1
Flexion Hip Right, Cervical Flexion; Sagittal Plane	1	1
Flexion Hip Left, Cervical Flexion; Sagittal Plane	1	1

**Table 8 ijerph-17-04845-t008:** Items that have not meet the minimum set value for the degree of definition.

Variable	First Definition	Second Definition
**OHS**		
Excess thoracic kyphosis; sagittal plane	In the thoracic spine there appears a greater convexity, which increases the dorsal curvature and appears in dorsal hypokyphosis, seen from the sagittal plane	The convexity of the dorsal curve increases excessively during the movement, seen from the plane sagittal
**SM**		
Winged scapula, right arm flexion and left external shoulder rotation; back plane	When the left arm, in the external shoulder rotation, looks for the right hand that makes the internal shoulder rotate and the lower peak of the right scapula is shown, viewed from the front/back plane	Pronunciation of the right scapular peak when the left arm is in external shoulder rotation and the right is in internal rotation, seen from the front/back plane
Winged scapula, left arm flexion and right external shoulder rotation; back plane	When the right arm, in the external shoulder rotation, looks for the left hand that makes the internal shoulder rotate and the lower peak of the left scapula is shown, viewed from the front/back plane.	Pronunciation of the left scapular peak when the right arm is in external shoulder rotation and the left is in internal rotation, seen from the front/back plane
Excess lumbar lordosis, right arm flexion and left external shoulder rotation; sagittal plane	When the movement is performed with the left arm in shoulder flexion and external rotation, it looks for the left hand and the lumbar area shows hyper lordosis, seen from the sagittal plane.	The concavity of the lumbar curve increases excessively during the movement when the left arm is in external rotation and the right is in internal rotation, seen from the sagittal plane.
Excess lumbar lordosis, left arm flexion and right external shoulder rotation; sagittal plane	When the movement is performed with the right arm in shoulder flexion and external rotation, it looks for the left hand and the lumbar area shows hyper lordosis, seen from the sagittal plane.	The concavity of the lumbar curve increases excessively during the movement when the right arm is in external rotation and the left is in internal rotation, seen from the sagittal plane
Cervical protraction, right arm flexion and left external shoulder rotation; sagittal plane	When the movement is performed with the left arm in shoulder flexion and external rotation, it looks for the right hand and cervical ante pulsion appears, seen from the sagittal plane	The pterygoid vertical line is more advanced than at the start of motion when the left arm is in external rotation and the right in internal rotation, seen from the sagittal plane
Cervical protraction, left arm flexion and right external shoulder rotation; sagittal plane	When the movement is performed with the right arm in shoulder flexion and external rotation, it looks for the left hand and cervical ante pulsion appears, seen from the sagittal plane	The pterygoid vertical line is more advanced than at the start of motion when the right arm is in external rotation and the left in internal rotation, seen from the sagittal plane

**Table 9 ijerph-17-04845-t009:** Intra-observer reliability, Kappa value.

Observer	OHST	HST	FSDT	SMT	ASLR	VAFB
Observer 1	0.80	0.67	0.69	1	0.69	0.73
Observer 2	0.60	0.67	0.22	1	0.35	0.49
Observer 3	0.12	0.19	0.02	0.67	0.24	0.18
Observer 4	0.42	0.54	0.02	0.57	0.43	0.36
Observer 5	0.25	−0.30	0.62	0.40	0.74	0.52
Observer 6	0.69	0.24	0.24	1	0.13	0.27

**Table 10 ijerph-17-04845-t010:** Inter-observer reliability, Kappa value.

Observers	1st Observation	2nd Observation
Observers 1–2	0.46	0.58
Observers 1–3	0.29	0.07
Observers 1–4	0.36	0.56
Observers 1–5	0.35	0.18
Observers 1–6	0.50	0.17
Observers 2–3	0.24	0.12
Observers 2–4	0.44	0.30
Observers 2–5	0.38	0.21
Observers 2–6	0.44	0.11
Observers 3–4	0.13	0.17
Observers 3–5	0.25	0.15
Observers 3–6	0.29	0.17
Observers 4–5	0.34	0.13
Observers 4–6	0.31	0.14
Observers 5–6	0.39	0.23

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
