# Peer review of "Validity and Reliability of the New Basic Functional Assessment Protocol (BFA)"

_ijerph, 2020, doi:10.3390/ijerph17134845_

Round 1
Reviewer 1 Report
I enjoy the reading.
Comments only regarding the paper presentation and format.
Long tables (i.e. table 3) are difficult to be read. Please format them better to be easily read and understood. (maybe sub table could make the data more readable). I would split and spread the figures along with the paper instead of placing all of them at the end of the paper.
Author Response
Thank you very much for the comments and suggestions proposed for the article “Validity and reliability of new the Basic Functional Assessment protocol (BFA)”.
First of all, thanks for the useful suggestions and the opportunity to improve this document.
We are pleased to provide this letter to explain, point by point, the details of the revisions and modifications made to the manuscript and our responses to the comments. Any review is "clearly highlighted" by controlling changes within the manuscript.
We want to confirm that an exhaustive review has been carried out in accordance with the proposed considerations. The changes made are explained after reviewers comments.
Detailed explanation point by point:
- Comment 1: I enjoy the reading.
Answer 1: We are very pleased that the reading of the article has been of your liking just as the authors have enjoyed during its development.
Comments only on the presentation and format of the article
- Comment 2: Long tables (i.e. table 3) are difficult to be read. Please format them better to be easily read and understood. (maybe sub table could make the data more readable)
Answer 2: The format of Table 3 has been modified, with the intention of making it easier to read and understand. Line: 240-258.
- Comment 3: I would split and spread the figures along with the paper instead of placing all of them at the end of the paper.
Answer 3: The figures have been included so that they are seen in the paper, maintaining them also at the end of the paper. Line: 228.

Reviewer 2 Report
In this manuscript, the authors aimed to develop a novel assessment tool for the detection of injury risks. They asked eight qualified judges to evaluate the validity of the tool and six external observers to perform an inter-intra reliability assessment. The merit of this manuscript is that the authors organized a pool of definitions and graphic images for their assessment protocol and gathered evaluation information from eight qualified judges, which may provide a valuable source for future investigations. The limitation of this manuscript is that they do not have any field-based measures that can evaluate the actual risk (or criterion validity).
Major concerns:
1), To my knowledge, Cohen’s kappa is used to measure reliability for qualitative data. However, in the current manuscript, the test scores might be bettered considered quantitative data, do the authors have any comment on this?
2), Can the authors provide more information regarding the characteristics of the judges? E.g., age, gender, what kind of degree, the average years of experience, etc.
3), Grammatical errors and typos.
Author Response
Thank you very much for the comments and suggestions proposed for the article “Validity and reliability of new the Basic Functional Assessment protocol (BFA)”.
First of all, thanks for the useful suggestions and the opportunity to improve this document.
We are pleased to provide this letter to explain, point by point, the details of the revisions and modifications made to the manuscript and our responses to the comments. Any review is "clearly highlighted" by controlling changes within the manuscript.
We want to confirm that an exhaustive review has been carried out in accordance with the proposed considerations. The changes made are explained after reviewers comments.
Detailed explanation point by point:
Comment 1: To my knowledge, Cohen’s kappa is used to measure reliability for qualitative data. However, in the current manuscript, the test scores might be bettered considered quantitative data, do the authors have any comment on this?
Answer 1: The data evaluated using Cohen's Kappa value are qualitative data “knee valgus, pelvic tilt, presence of kyphosis…” The test score would be as follows: if after the assessment of the subject, one of the different qualitative data appears in the result “knee valgus, pelvic tilt, presence of kyphosis…” There must be counted “1” for every fact that should appear and “0” if it does not appear. This would result in the sum of how many "qualitative data" the subject presents in its assessment.
Comment 2: Can the authors provide more information regarding the characteristics of the judges? E.g., age, gender, what kind of degree, the average years of experience, etc.
Answer 2: The Information on the characteristics of judges has been expanded. Similarly, information on the six external observers has been expanded. Line: 173.
Comment 3: Grammatical errors and typos
Answer 3: The document has been revised completely to avoid the presence of grammatical and typographical errors. Line: 68,76, 333,360…

Reviewer 3 Report
The introduction does not include enough references to contextualize what is being studied.
The method should be more detailed in the validation procedure.
In addition, reference 36 appears twice on line 111-112: Shoulder Mobility (SM) [36] and Active Elevation of Stretched Legs 112 (ASLR) [36]. Is this a mistake?
In the results section, authors provide many tables but do not comment in depth.
The discussion section is well structured.
The conclusion section is quite weak. The main research findings, study limitations and future lines of research should be included.
Author Response
Thank you very much for the comments and suggestions proposed for the article “Validity and reliability of new the Basic Functional Assessment protocol (BFA)”.
First of all, thanks for the useful suggestions and the opportunity to improve this document.
We are pleased to provide this letter to explain, point by point, the details of the revisions and modifications made to the manuscript and our responses to the comments. Any review is "clearly highlighted" by controlling changes within the manuscript.
We want to confirm that an exhaustive review has been carried out in accordance with the proposed considerations. The changes made are explained after reviewers comments.
Detailed explanation point by point:
- Comment 1: The introduction does not include enough references to contextualize what is being studied.
Answer 1: The information has been expanded in the introduction, with the intention of providing better contextualization of the study.Line: 48, 58-63.
- Comment 2: The method should be more detailed in the validation procedure.
Answer 2: More detailed data on the validation procedure has been included. Line 331-148.
- Comment 3: In addition, reference 36 appears twice on line 111-112: Shoulder Mobility (SM) [36] and Active Elevation of Stretched Legs 112 (ASLR) [36]. Is this a mistake?
Answer 3: This is not an error, both tests are taken from that reference. Since it is the FMS battery.
- Comment 4: In the results section, authors provide many tables but do not comment in depth.
Answer 4: The comments on results have been expanded. Line: 216- 227.
- Comment 5: The discussion section is well structured.
Answer 5: The discussion maintains its structure and information. Line: 283.
- Comment 6: The conclusion section is quite weak. The main research findings, study limitations and future lines of research should be included.
Answer 6: The information in the conclusion has been strengthened. Line 412.

Round 2
Reviewer 2 Report
Thank the authors for making the revisions.
Reviewer 3 Report
The manuscript has improved since the last revision. Authors have made the suggested corrections.